# Stop Unnecessary Reflection: Training LRMs for Efficient Reasoning with Adaptive Reflection and Length Coordinated Penalty

**Zewei Yu**[1,2]    **Lirong Gao**[1,2]    **Yuke Zhu**[2]    **Bo Zheng**[2]    **Junbo Zhao**[1]
**Sheng Guo**[2*]  **Haobo Wang**[1,3*]
[1] State Key Laboratory of Blockchain and Data Security, Zhejiang University
[2] MYbank, Ant Group
[3] Innovation and Management Center, School of Software Technology (Ningbo), Zhejiang University
{yuzeweizju, gaolirong, j.zhao, wanghaobo}@zju.edu.cn
felix.yk@alibaba-inc.com  guangyuan@antgroup.com
guosheng1001@gmail.com

## Abstract

Large Reasoning Models (LRMs) have demonstrated remarkable performance on complex reasoning tasks by employing test-time scaling. However, they often generate over-long chains-of-thought that, driven by substantial reflections such as repetitive self-questioning and circular reasoning, lead to high token consumption, substantial computational overhead, and increased latency without improving accuracy, particularly in smaller models. Our observation reveals that increasing problem complexity induces more excessive and unnecessary reflection, which in turn reduces accuracy and increases token overhead. To address this challenge, we propose **A**daptive **R**eflection and **L**ength **C**oordinated **P**enalty (**ARLCP**), a novel reinforcement learning framework designed to dynamically balance reasoning efficiency and solution accuracy. ARLCP introduces two key innovations: (1) a reflection penalty that adaptively curtails unnecessary reflective steps while preserving essential reasoning, and (2) a length penalty calibrated to the estimated complexity of the problem. By coordinating these penalties, ARLCP encourages the model to generate more concise and effective reasoning paths. We evaluate our method on five mathematical reasoning benchmarks using DeepSeek-R1-Distill-Qwen-1.5B and DeepSeek-R1-Distill-Qwen-7B models. Experimental results show that ARLCP achieves a superior efficiency-accuracy trade-off compared to existing approaches. For the 1.5B model, it reduces the average response length by 53.1% while simultaneously improving accuracy by 5.8%. For the 7B model, it achieves a 35.0% reduction in length with a 2.7% accuracy gain. The code is released at https://github.com/ZeweiYu1/ARLCP.

## 1 Introduction

Large Reasoning Models (LRMs), such as OpenAI o1 (OpenAI, 2024), QwQ (Qwen Team, 2025), and DeepSeek-R1 (Guo et al., 2025), have demonstrated exceptional capabilities in complex reasoning tasks. When tackling challenging problems, these models employ long chain-of-thought reasoning with self-reflective mechanisms, systematically exploring multiple solution pathways while generating extensive reflective reasoning traces. This iterative reasoning framework enables them to perform significantly better than conventional LLMs. Typically, the reasoning process in these LRMs is explicitly organized with `<think>` and `</think>` tags, which separate internal thinking from final outputs, offering transparent insight into its multi-step reasoning process.

Although overlong reasoning improves the performance of LRMs, it also brings a severe efficiency challenge with substantial token usage and computational overhead, limiting practical deployment in real-time or resource-constrained settings (Qu et al., 2025; Yue et al., 2025; Yang et al., 2025c).

---
*Corresponding Authors

Existing research has mainly explored two directions to address this efficiency challenge. The first is training-free inference-stage optimization, such as Early Exit (Yang et al., 2025b; Qiao et al., 2025; Xu et al., 2025) and Model Switch (Liao et al., 2025; Ong et al., 2025; Yang et al., 2025c). These methods do not change the model's inference capability or distribution, but only optimize the generation process with early stopping or pruning, thus offering limited efficiency gains for redundant models or complex tasks. The second direction is training LRMs under length-penalty guidance using supervised fine-tuning (Jiang et al., 2025; Xia et al., 2025; Huang et al., 2025) or Reinforcement Learning (RL) (Arora & Zanette, 2025; Zhang et al., 2025; Aggarwal & Welleck, 2025). While such methods enhance control over reasoning length, they often sacrifice reasoning quality—for example, by suppressing reflection or discarding the entire thinking process—which negatively affects answer accuracy. Therefore, dynamically balancing length and accuracy remains a valuable direction for achieving efficient reasoning.

Based on detailed analysis of the reasoning process, we find that LRMs, particularly smaller ones, frequently produce redundant reasoning steps, such as repetitive self-questioning loops ("wait"), unproductive hesitations ("hmm"), and circular reflections that fail to advance task resolution, as also evidenced in (Yang et al., 2025c; Ghosal et al., 2025). Furthermore, we conduct an in-depth analysis of the reflective behaviors of LRMs during the reasoning process. Defining **complexity as a model-aware measure of problem difficulty**, we observed (as shown in Figure 1 and Figure 2) that **as the complexity of the problem increases, this over-reflection phenomenon becomes more severe**, incurring significant costs in inference time and computation. More critically, even after prolonged over-reflection during the reasoning process, LLMs still fail to yield correct answers.

Motivated by these observations, we propose **A**daptive **R**eflection and **L**ength **C**oordinated **P**enalty (**ARLCP**), a dynamic reinforcement learning method that enables LRMs to balance accuracy and efficiency in reasoning. The core of ARLCP is to reduce unnecessary reflection behaviors of LRMs during reasoning while preserving accuracy, thereby reducing token consumption and improving efficiency. Unlike prior approaches that naively truncate reasoning steps or apply static penalties, ARLCP introduces two key innovations: (1) a reflection penalty that adaptively stops unnecessary reflection while preserving critical reflection processes, and (2) a length penalty that reduces the output tokens, calibrated to problem complexity. By dynamically adjusting reflection tokens, ARLCP automatically adjusts penalty weights to maintain accuracy while minimizing token overhead.

We evaluate ARLCP on five mathematical reasoning benchmarks using DeepSeek-R1-Distill-Qwen-1.5B and DeepSeek-R1-Distill-Qwen-7B as the base models. The experimental results demonstrate that ARLCP achieves significant efficiency-accuracy improvements: on the 1.5B model, it reduces average response length by 53.1% while improving accuracy by 5.8%, and on the 7B model, it achieves a 35.0% response-length reduction with a 2.7% accuracy improvement. This consistent performance across different model scales validates the effectiveness of our approach in balancing reasoning efficiency and solution quality.

In summary, our key contributions are as follows:

- We identify the phenomenon of over-reflection in reasoning models, where redundant or unproductive reasoning steps degrade inference efficiency and repurpose such signals inversely to terminate unnecessary reasoning proactively.
- We introduce **ARLCP**, a reinforcement learning method that dynamically adjusts reflection and length penalties based on online complexity estimates.
- We evaluate ARLCP across multiple math datasets, demonstrating superior performance-efficiency trade-offs: our method significantly reduces token consumption while preserving or improving accuracy, outperforming existing approaches for efficient reasoning.

## 2 RELATED WORK

**Large Reasoning Models.** Following Open-o1 (OpenAI, 2024), researchers have developed advanced reasoning models through detailed rewards and search-based methods (Qwen Team, 2025). Notable approaches include mutual learning between models (Qi et al., 2024), example-guided search (Zhang et al., 2024), and MCTS-integrated self-play for self-correcting reasoning (Zhao et al., 2024). The release of DeepSeek-R1 (Guo et al., 2025) further popularized "R1-style" models that achieve multi-step reasoning and self-reflection using only simple rule-based rewards (Team et al.,

2025; Yang et al., 2025a). However, overthinking behaviors significantly increase computational costs, driving active research into efficiency reasoning.

**Efficient Reasoning for LRMs.** Most existing methods to improve the efficiency of LRM focus on reducing response tokens. Training-free approaches include prompting with token budgets (Muennighoff et al., 2025; Aytes et al., 2025), model switching (Liao et al., 2025; Fan et al., 2025; Yang et al., 2025c), and early exit mechanisms (Yang et al., 2025b; Qiao et al., 2025). Other strategies use Supervised Fine-Tuning (SFT) with compressed Chain-of-Thought (CoT) data (Jiang et al., 2025; Yu et al., 2025) or length-selected data from sampling and post-processing (Shen et al., 2025; Rafailov et al., 2023). Furthermore, Reinforcement Learning (RL) techniques often incorporate length-based rewards (Arora & Zanette, 2025; Luo et al., 2025a; Liu et al., 2025a) or adapt other rewards (Zhang et al., 2025; Aggarwal & Welleck, 2025). While different, these methods fail to dynamically adapt response length to a problem's intrinsic complexity or mitigate a model's over-reflection tendencies. Our work addresses this gap by introducing a synergistic penalty mechanism that incorporates reflection and length penalties to achieve efficient reasoning.

## 3 MOTIVATION AND OBSERVATION

This section examines common patterns that regularly appear during models' reasoning processes. By carefully studying these patterns, we hope to find practical mechanisms to improve the efficiency of the models.

**Reflection correlates with problem complexity.**
During inference, we observe that reasoning models frequently generate specific reasoning-supportive tokens such as "wait", "hmm", and "alternatively", which are intrinsically linked to the model's internal behavior. To quantitatively analyze these patterns, we conduct a study measuring average reflection token counts across Deepseek-Distilled-Qwen-1.5B and 7B on multiple math datasets with an increased complexity level, as shown in Figure 1. Notably, the number of reflection tokens positively correlates with dataset complexity, demonstrating their sensitivity to problem complexity.

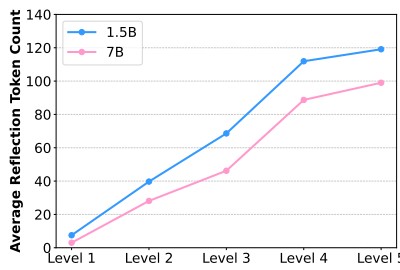

Figure 1: Average reflection token counts statistics.

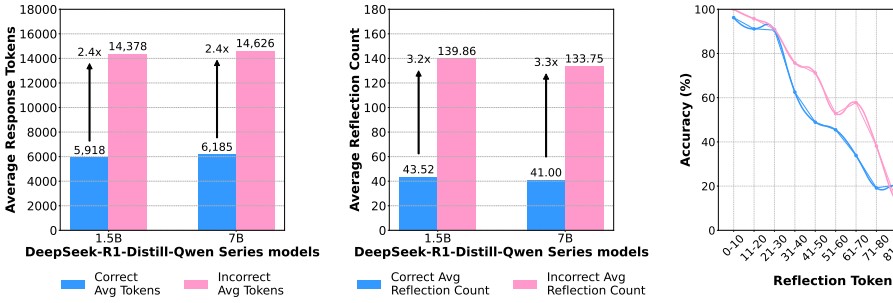

Figure 2: Output statistics of two models on the AIME 2024-2025 datasets. The reported metrics include average response tokens (left), average reflection token count (medium) for both correct and incorrect answers, and accuracy trend with different reflection token count intervals.

**Over-reflection leads to incorrect responses and inefficiency.** To further investigate the relationship between reflection tokens and model performance, we analyze the performance of DeepSeek-R1-Distilled-Qwen-1.5B and 7B in terms of response length and reflection tokens for both correct answers and incorrect answers and the accuracy trend with different reflection token count intervals on AIME 2024-2025 datasets, as shown in Figure 2. The results indicate that correct responses

exhibit significantly shorter average token lengths and fewer reflection tokens than incorrect ones. And smaller reasoning models (1.5B) require substantially longer responses while exhibiting lower accuracy than larger models (7B). Additionally, as the number of reflection tokens increases, model accuracy declines. These results indicate that while some reflection is necessary, unproductive or excessive reflection—a phenomenon we term "over-reflection"—traps models in inefficient and often incorrect exploration of the solution space.

**Key insights and motivation.**    Our observations highlight three key insights:

- **Reflection as complexity indicator:** Higher reflective token counts correlate with increased problem complexity, enabling dynamic assessment and adaptive reward design.
- **Over-reflection risks:** Incorrect responses often exhibit excessive reflection, trapping models in inefficient solution-space exploration.
- **Balanced intervention needed:** Low-complexity tasks still require length penalties for their low-reflection behaviors, while complex tasks require strategies to stop unnecessary reflections.

Motivated by these observations, we propose ARLCP, a reinforcement learning method that combines two key components: (1) a dynamic reflection penalty adjusted based on problem complexity and (2) a length penalty to ensure an overall penalty. By balancing these two penalties, our approach systematically reduces token usage while maintaining or improving model accuracy through optimized reasoning processes.

## 4    METHODOLOGY

In this section, we introduce **A**daptive **R**eflection and **L**ength **C**oordinated **P**enalty (**ARLCP**) algorithm, which consists of two essential components: (1) an adaptive reflection penalty based on the complexity of problems. (2) a length penalty ensures the overall penalty in a low-reflection situation.

### 4.1    PRELIMINARY

Given an LRM $M$, an input prompt $x = [x_1, \ldots, x_n, \texttt{<think>}]$, where $[x_1, \ldots, x_n]$ represents the problem context and $\texttt{<think>}$ serves as the special token initiating the reasoning process, $M$ generates a response $y = [y_1, \ldots, y_l, \texttt{</think>}, y_{l+2}, \ldots, y_m]$. Here, $[y_1, \ldots, y_l]$ represents the thinking phase, constituting an extended chain of exploratory reasoning, reflection, and self-validation. while $\texttt{</think>}$ explicitly terminates this process. $[y_{l+2}, \ldots, y_m]$ is the subsequent solution segment which contains only the validated steps and final answer to the problem. Since the generation is auto-regressive, meaning that given a prompt $x$ and tokens $y^{\leq k} = [y_1, \ldots, y_k]$ generated so far, next token $y^{k+1}$ is generated from the conditional probablity distribution $M(y_{k+1} \mid x, y_{\leq k})$. The whole auto-regressive steps can be decomposed as:

$$M(y|x) = \prod_{t=k}^{m} M(y_k|x, y_{<k}). \tag{1}$$

The auto-regressive generation process terminates when the reasoning model $M$ outputs the end-of-sequence (EOS) token. Therefore, in the response $y = [y_1, \ldots, y_l, \texttt{</think>}, y_{l+2}, \ldots, y_m]$, the final token $y_m$ always corresponds to the EOS token.

### 4.2    ADAPTIVE REFLECTION AND LENGTH COORDINATED PENALTY (ARLCP)

Our proposed approach ARLCP is a reinforcement learning method that adaptively imposes a reflection penalty according to the complexity of each problem, supplemented by a length penalty, allowing the LRM to flexibly reduce unnecessary reflection and thus minimize token consumption. Based on empirical observations in Section 3, we identify three key findings: (1) reflection intensity correlates with problem complexity; (2) incorrect solutions exhibit excessive reflection patterns that hinder efficient reasoning; (3) static reflection penalty is ineffective for low-reflection problems.

**The complexity of each problem $\lambda$.**    Motivated by the above findings, we first estimate the model-aware complexity of each problem based on the Reflection Token Counts (RTC) of the

---

**Algorithm 1** *ARLCP (Adaptive Reflection and Length Coordinated Penalty)*

---

**Input:** Policy model $M$; dataset $\mathcal{D}$; hyperparameters $m, \alpha, n_1, n_2, \lambda_1, \lambda_2, \lambda_3$

---

1: **Initialize:** Ground truth answers $o^*(P_i)$ for all $P_i \in \mathcal{D}$
2: **for** training step $= 1, \ldots, M$ **do**
3:      Sample a batch $\mathcal{D}_b$ from $\mathcal{D}$
4:      For each $P_i \in \mathcal{D}_b$, generate $m$ candidate responses $o_i^k \sim M(\cdot|P_i)$ using sampling
5:      **for all** $o_i^k$ in batch **do**
6:          Extract $\text{LEN}(o_i^k)$ (response token count), $\text{RTC}(o_i^k)$ (reflection token count via keyword matching),
             and $\text{ANS}(o_i^k)$
7:          Compute correctness $\mathcal{C}(o_i^k) \leftarrow \mathbf{1}\{\text{ANS}(o_i^k) = o^*(P_i)\}$
8:      **end for**
9:      Collect correct responses $\mathcal{Y}_{\text{correct}} = \{o_i^k | \mathcal{C}(o_i^k) = 1\}$
10:     Compute $\mu_R = \text{mean}(\text{RTC}(\mathcal{Y}_{\text{correct}})), \sigma_R = \text{std}(\text{RTC}(\mathcal{Y}_{\text{correct}}))$
11:     Compute $\mu_L = \text{mean}(\text{LEN}(\mathcal{Y}_{\text{correct}})), \sigma_L = \text{std}(\text{LEN}(\mathcal{Y}_{\text{correct}}))$
12:     **for all** $o_i^k$ in batch **do**
13:        Normalize penalties: $f(\text{RTC}) = \sigma\left(\frac{\text{RTC}(o_i^k) - \mu_R}{\sigma_R}\right), f(\text{LEN}) = \sigma\left(\frac{\text{LEN}(o_i^k) - \mu_L}{\sigma_L}\right)$
14:        Determine $\alpha_1$ via thresholding: $\alpha_1 = \begin{cases} \lambda_1, & \text{if } \text{RTC}(o_i^k) \leq n_1 \\ \lambda_2, & \text{if } n_1 < \text{RTC}(o_i^k) \leq n_2 \\ \lambda_3, & \text{if } \text{RTC}(o_i^k) > n_2 \end{cases}$
15:        Compute $\alpha_2 = \alpha - \alpha_1$
16:        Calculate reward $r(o_i^k) = \mathcal{C}(o_i^k) \cdot (1 - \alpha_1 f(\text{RTC}) - \alpha_2 f(\text{LEN}))$
17:      **end for**
18:      Update $M$ using policy gradient update
19: **end for**

**Output:** Optimized policy $M$ with adaptive penalty control

---

response. Specifically, for each prompt $p_i$ in the batch, the LRM generates $m$ candidate rollouts $o_i = [o_i^1, o_i^2 \ldots, o_i^m]$ using standard sampling. Each reasoning trajectory $o_i^k$ consists of: (1) $\text{LEN}(o_i^k)$, indicating the response length tokens. (2) $\text{RTC}(o_i^k)$, indicating the reflection tokens count, based on reflection-trigger keyword matching, as shown in Appendix A.4. The complexity of the problem $\lambda$ is categorized into three levels: *simple*, *moderate*, and *hard* through threshold-based segmentation: *simple* with weight $\lambda_1$ when $\text{RTC}(o_i^k)$ does not exceed the lowest threshold $n_1$, *moderate* with weight $\lambda_2$ when $\text{RTC}(o_i^k)$ falls between $n_1$ and the moderate threshold $n_2$, and *hard* with weight $\lambda_3$ when $\text{RTC}(o_i^k)$ exceeds the highest threshold $n_2$.

**Reflection penalty.** Then, we propose an adaptive mechanism that dynamically imposes a reflection penalty according to the complexity of each problem $\lambda$. This mechanism allows the LRM to flexibly adjust its reasoning process, discouraging unnecessary reflection on simpler problems while permitting more extensive reasoning for complex ones. Formally, we define the reflection penalty coefficient $\alpha_1$, which is modulated by the estimated problem complexity $\lambda$, as follows:

$$\alpha_1 = \begin{cases} \lambda_1, & \text{if } \text{RTC}(o_i^k) \leq n_1; \\ \lambda_2, & \text{if } n_1 < \text{RTC}(o_i^k) \leq n_2; \\ \lambda_3, & \text{if } \text{RTC}(o_i^k) > n_2. \end{cases} \tag{2}$$

For reasoning trajectory $o_i^k$, its reflection penalty $f(\text{RTC}(o_i^k))$ can be calculated as:

$$f(\text{RTC}(o_i^k)) = \sigma\left(\frac{\text{RTC}(o_i^k) - \text{mean}(\text{RTC}(o_i))_{\text{correct}}}{\text{std}(\text{RTC}(o_i))_{\text{correct}}}\right). \tag{3}$$

Here, $\text{mean}(\text{RTC}(o_i))_{\text{correct}}$ and $\text{std}(\text{RTC}(o_i))_{\text{correct}}$ are the mean and standard deviation of reflection tokens whose answers are correct, respectively. $\sigma$ is the sigmoid function.

**Length penalty.** While the reflection penalty effectively discourages excessive reflective behaviors, it may not fully constrain other forms of unnecessary verbosity that do not directly stem from reflection. Therefore, we further introduce a length penalty based on the total token count $\text{LEN}(o_i^k)$

of the generated output. This complementary penalty encourages the LRM to generate overall more concise responses, suppressing both redundant reflection and any additional irrelevant or verbose content. The length penalty is calculated as:

$$f(\text{LEN}(o_i^k)) = \sigma\left(\frac{\text{LEN}(o_i^k) - \text{mean}(\text{LEN}(o_i))_{\text{correct}}}{\text{std}(\text{LEN}(o_i))_{\text{correct}}}\right). \tag{4}$$

The length penalty coefficient $\alpha_2$ of $o_i^k$ is defined as $\alpha_2 = \alpha - \alpha_1$, where $\alpha$ is the overall penalty coefficient. This coefficient allows the total penalty to be flexibly allocated between reflection and length penalties according to the complexity of each problem.

Building on the above, for each rollout $o_i^k$, the composite reward function jointly optimizes for accuracy and efficiency, and is formulated as follows:

$$r(o_i^k) = \mathcal{C}(o_i^k) \cdot (1 - \alpha_1 f(\text{RTC}(o_i^k)) - \alpha_2 f(\text{LEN}(o_i^k))), \tag{5}$$

where the $\text{ANS}(o_i^k)$ represents the extracted answer from the solution segment and $o^*(p_i)$ be the ground truth of $p_i$. Hence, the accuracy reward $\mathcal{C}(o_i^k)$ is calculated by:

$$\mathcal{C}(o_i^k) = \mathbf{1}\{\text{ANS}(o_i^k) = o^*(p_i)\}. \tag{6}$$

The adaptive penalty strategy maintains accuracy while effectively stopping unnecessary reflections and reducing the response tokens. During each training step, the LRM samples $m$ rollouts $o_i = [o_i^1, o_i^2 \ldots, o_i^m]$. Once sampling finishes, correct responses are calculated to compute $\text{mean}(\text{RTC}(o_i))_{\text{correct}}$, $\text{std}(\text{RTC}(o_i))_{\text{correct}}$, $\text{mean}(\text{LEN}(o_i))_{\text{correct}}$, and $\text{std}(\text{LEN}(o_i))_{\text{correct}}$. Accuracy rewards, reflection penalties, and length penalties are then calculated for each response, while coefficients $\alpha_1$ and $\alpha_2$ are determined based on the response complexity. The overall penalty is computed accordingly. Finally, the advantage of each response is estimated using online RL algorithms (leveraging multiple responses) for policy gradient update.

## 5 EXPERIMENTS

### 5.1 EXPERIMENTAL SETUP

**LRMs.** We conduct all experiments on DeepSeek-R1-Distill-Qwen-1.5B and DeepSeek-R1-Distill-Qwen-7B, two popular reasoning models obtained through supervised fine-tuning on large-scale high quality distillation.

**Datasets.** The training dataset we use is DeepScaleR (Luo et al., 2025b) dataset, which is a collection of 40K unique math problem-answer pairs compiled from: AIME problems (1984-2023), AMC (prior to 2023), Omni-MATH (Gao et al., 2025), and Still dataset (Min et al., 2024). The training dataset does not overlap with our evaluation benchmark dataset. For evaluation, we use five widely recognized math benchmarks with increasing complexity: GSM8K (Cobbe et al., 2021) test set (1319 grade school math problems), MATH500 (Lightman et al., 2024) (500 high-school competition math problems), AMC2023 (40 more challenging high-school level math competition problems), AIME 2024 and AIME 2025 (30 Olympiad-level math problems).

**Evaluation metrics** For evaluation metrics, we consider both pass@1 accuracy (Acc) and response length (Length). We also report the average accuracy variation, $\Delta Acc$, and the average length variation rate, $\Delta Length(\%)$, across all test datasets. Considering the limited size of AMC2023, AIME2024, and AIME2025, we repeatedly sample 16 responses for each case and report the average results. For all models, we set the evaluation context size to 16K, and set the temperature to 0.6 as default in DeepSeek's models.

**Implementation Details.** Our implementation integrates the VeRL framework (Sheng et al., 2025) with REINFORCE Leave One Out (RLOO) policy optimization method (Ahmadian et al., 2024). While GRPO (Shao et al., 2024) has gained popularity as a policy optimization approach, we avoid its use due to two critical limitations: (1) GRPO demonstrates sensitivity in non-standard settings involving length penalties in the objective function, and (2) this sensitivity can trigger abrupt policy collapses, as empirically observed in recent studies (Arora & Zanette, 2025; Dai et al., 2025). These

Table 1: Performance comparison of different methods on multiple reasoning benchmarks. Metrics include accuracy (Acc), response tokens (Length), and relative changes ($\Delta$Acc, $\Delta$Length). The best and second results are bolded and underlined, respectively.

| Method | AMC 2023 | | AIME 2024 | | AIME 2025 | | GSM8K | | MATH-500 | | Overall | |
| | Acc | Length | Acc | Length | Acc | Length | Acc | Length | Acc | Length | $\Delta$Acc | $\Delta$Length(%) |
|---|---|---|---|---|---|---|---|---|---|---|---|---|
| *DeepSeek-R1-Distill-Qwen-1.5B* | | | | | | | | | | | | |
| Vanilla | 66.72 | 7742 | 30.00 | 12256 | 21.40 | 12201 | 78.46 | 1009 | 80.20 | 4582 | - | - |
| NoThinking | 49.22 | 1377 | 14.38 | 4266 | 9.79 | 3630 | 69.98 | 278 | 69.20 | 908 | -12.84 | -81.04% |
| SFT$_{Shortest}$ | 67.66 | 7657 | 26.04 | 12181 | 21.46 | 12258 | 79.38 | 928 | 81.00 | 4531 | -0.25 | -2.07% |
| DPO$_{Shortest}$ | 69.53 | 7091 | 29.60 | 11721 | 23.13 | 11879 | 77.10 | 989 | 84.20 | 4185 | 1.36 | -5.21% |
| O1-Pruner | 70.47 | 7046 | 27.71 | 11937 | 20.83 | 11855 | 78.39 | 913 | 82.60 | 4375 | 0.64 | -5.69% |
| TLMRE | 72.10 | **2798** | 25.80 | **5386** | 19.60 | **4581** | 84.30 | 530 | 82.10 | **1800** | 1.42 | **-58.10%** |
| AdaptThink | 67.19 | 3342 | 30.83 | 6864 | 22.50 | 6626 | 84.23 | **488** | 83.20 | 1869 | 2.23 | -51.47% |
| LASER | **75.94** | 3899 | 28.75 | 7629 | 25.42 | 7084 | 82.26 | 850 | **84.60** | 2365 | 4.04 | -38.69% |
| **ARLCP(ours)** | 73.28 | 3037 | **34.17** | 5951 | **26.46** | 5196 | **87.34** | 658 | **84.60** | 1812 | **5.81** | -53.05% |
| *DeepSeek-R1-Distill-Qwen-7B* | | | | | | | | | | | | |
| Vanilla | 87.50 | 5980 | 51.46 | 10547 | 35.00 | 11427 | 87.34 | 694 | **91.20** | 3673 | - | - |
| Nothinking | 61.25 | 1179 | 23.96 | 2671 | 16.25 | 2499 | 84.68 | 284 | 80.60 | 706 | -17.21 | -74.59% |
| SFT$_{Shortest}$ | 85.80 | 6005 | 49.17 | 10575 | 38.96 | 11384 | 86.43 | 606 | 89.00 | 3575 | -0.69 | -3.09% |
| DPO$_{Shortest}$ | 88.90 | 5295 | 50.42 | 9899 | **39.58** | 10692 | 87.87 | 624 | 90.80 | 3340 | 0.95 | -8.64% |
| O1-Pruner | 87.66 | 5509 | 51.46 | 10034 | 38.33 | 10793 | 86.80 | 625 | **91.20** | 3422 | 0.53 | -7.01% |
| TLMRE | 88.12 | 3560 | 52.08 | 6393 | 35.63 | 6584 | 88.78 | 866 | **91.20** | 2419 | 0.60 | -26.32% |
| AdaptThink | 86.88 | 3641 | 56.25 | 8784 | 37.71 | 9622 | **90.29** | **330** | 91.00 | **1862** | 1.87 | -34.68% |
| LASER | 89.45 | **2894** | 56.45 | **5995** | 39.38 | **6410** | 86.73 | 818 | 91.00 | 1871 | 2.04 | -33.97% |
| **ARLCP(ours)** | **89.69** | 3260 | **56.67** | 6795 | **39.58** | 7560 | 89.31 | 578 | 91.00 | 2086 | **2.69** | **-34.96%** |

instability risks motivate our choice of RLOO, which maintains robustness under length-penalized objectives. The details of selecting RLOO are shown in Appendix A.2. All experiments are conducted on 8 NVIDIA A100 GPUs with prompts using model-specific templates for problems (see Appendix A.6), context length of 16K tokens, batch size 128, learning rate 2e-6, and ARLCP parameters $m = 16, \lambda_1 = 0.05, \lambda_2 = 0.1, \lambda_3 = 0.15, n_1 = 40, n_2 = 80, \alpha = 0.2$. The details of selecting $n_1$ and $n_2$ are shown in Appendix A.7.

## 5.2 BASELINES

We compare ARLCP with the following representative methods for efficient reasoning:

1. **Nothinking** (Ma et al., 2025) enables reasoning models to bypass long thinking and directly generate the final solution by prompting with `</think>`. We use this method to determine the lower bound of the response token for each dataset and do not include it in the comparisons.

2. **SFT$_{Shortest}$** constructs training data by sampling multiple responses for each problem and selecting the two shortest correct responses, followed by the standard SFT pipeline for model fine-tuning.

3. **DPO$_{Shortest}$** generates preference data by pairing the shortest correct response with the longest responses per problem through multiple sampling, then applies DPO (Rafailov et al., 2023) for fine-tuning.

4. **O1-Pruner** (Luo et al., 2025a) employs pre-sampling to estimate reference model performance, followed by off-policy RL-style fine-tuning to optimize shorter reasoning processes while maintaining accuracy constraints.

5. **TLMRE** (Arora & Zanette, 2025) introduces a length-based penalty during on-policy RL training to explicitly incentivize shorter response generation.

6. **AdaptThink** (Zhang et al., 2025) enables reasoning models to adaptively select optimal thinking mode based on the problem complexity by on-policy RL.

7. **LASER** (Liu et al., 2025a) propose a unified view for RL-based CoT compression, unifying various reward shaping and truncation methods. Building on this view, they introduce new approaches with adaptive length-based reward shaping.

All baselines are re-implemented using DeepScaleR dataset to ensure fair comparative evaluation.

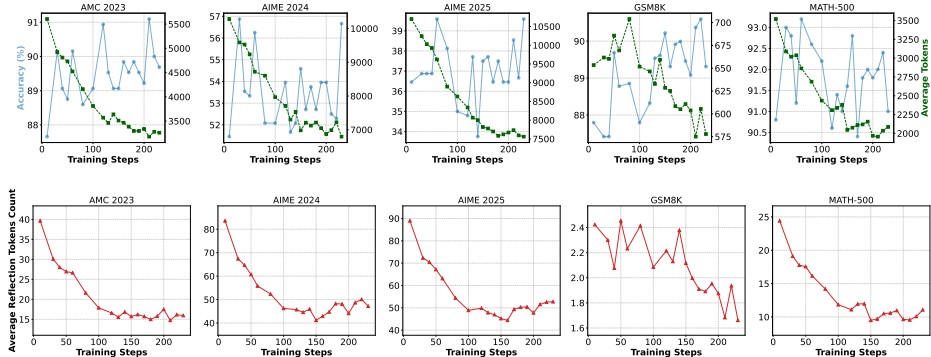

Figure 3: The analysis of accuracy, length, and reflection for model responses on five benchmarks (AMC 2023, AIME 2024, AIME 2025, GSM8K, and MATH 500) across different training steps.

## 5.3    MAIN RESULTS

Table 1 summarizes the evaluation results of various methods on five mathematical reasoning benchmarks (AMC 2023, AIME 2024, AIME 2025, GSM8K and MATH500). Compared to the vanilla DeepSeek-R1-Distill-Qwen-1.5B and DeepSeek-R1-Distill-Qwen-7B models, ARLCP achieves remarkable improvements by reducing average response lengths by 53.1% and 35.0%, respectively, while enhancing average accuracy by 5.8% and 2.7%. On **DeepSeek-R1-Distill-Qwen-1.5B**, ARLCP achieves the highest overall accuracy ($\Delta$**Acc = 5.81**) while reducing the average response length by **53.05%**. Similarly, for **DeepSeek-R1-Distill-Qwen-7B**, ARLCP maintains a significant accuracy gain ($\Delta$**Acc = 2.69**) and the greatest reduction in response length (**34.96%**). The experimental results of ARLCP across both DeepSeek-R1-Distill-Qwen-1.5B and DeepSeek-R1-Distill-Qwen-7B demonstrate a clear trade-off between reasoning accuracy and response length.

Furthermore, ARLCP excels particularly in complex tasks, attaining best results on competition-level benchmarks (**AMC 2023 (73.28%)**, **AIME 2024 (34.17%)**, **AIME 2025 (26.46%)**) for the 1.5B model. This suggests that, compared to simpler problems, LRMs tend to generate more unnecessary reflective behaviors when dealing with complex tasks. Thus, the key reason for the significant improvement lies in **adaptive reflection penalty mechanism**, which dynamically adjusts penalty strength according to problem difficulty, enabling more effective suppression of unnecessary reflections in complex tasks. ARLCP allows efficient reasoning without compromising solution quality, achieving the balance between accuracy and efficiency through coordinated penalty design.

To evaluate the effectiveness of our approach, we deeply track the changes in response length and the number of reflection-triggering words throughout the training process. As shown in Figure 3, both the average count of reflection words and the output length gradually decrease as training progresses, while accuracy correspondingly improves. These results indicate that our proposed adaptive reflection penalty successfully encourages the LRM to eliminate unnecessary reflective behaviors and produce more concise outputs. Consequently, this not only reduces superfluous token consumption but also significantly lowers inference costs, underscoring the practical value of our method for efficient large-scale deployment.

## 5.4    MORE ANALYSIS

**The analysis of reflection behavior.**    To evaluate the efficacy of our approach in mitigating unnecessary reflective behaviors, we conducted comparative analyses between ARLCP and the original models across multiple math datasets, as presented in Figure 4. The results demonstrate that our model significantly reduces the average reflection token count, effectively stopping unnecessary reflective behaviors in the model.

**The impact of two core penalty.**    To verify the cooperative effect of length penalty and reflection penalty in ARLCP, we conducted an ablation study on the 1.5B LRM by separately removing each component. When removing the reflection penalty, we fixed the length penalty parameter $\alpha_2$ to

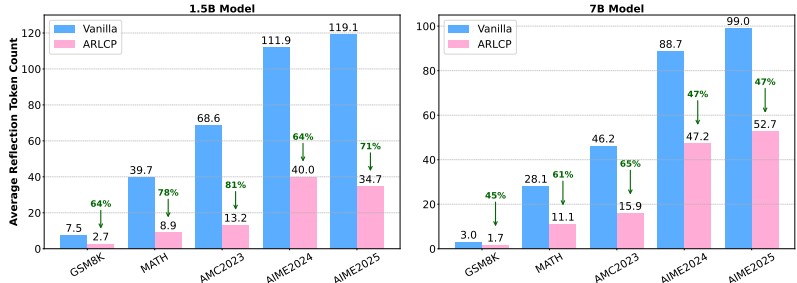

Figure 4: Comparison of average reflection token count between vanilla models and ARLCP.

Table 2: The impact of adaptive penalty components on the performance of LRM. The dynamic penalty yields the best balance between improved accuracy and reduced response length in LRM.

| Method | Length Penalty $\alpha_2$ | Reflection Penalty $\alpha_1$ | $\triangle$Acc | $\triangle$Length(%) |
|---|---|---|---|---|
| **ARLCP** | Adaptive(✓) | Adaptive(✓) | **5.8** | -53.1% |
| w/o Length Penalty | ✗ | ✓ | 5.6 | -29.6% |
| w/o Reflection Penalty | 0.05 | ✗ | 1.4 | **-58.1%** |
| Fixed Coefficients | 0.1 | 0.1 | 5.5 | -47.5% |

0.05; when removing the length penalty, we retained the original adaptive parameter $\alpha_1$. As shown in Table 2, removing either component led to performance degradation, confirming their cooperative roles in optimizing reasoning efficiency. We further analyze the impact of the adaptive coefficients on model performance. We fix the adaptive coefficients $\alpha_1$ and $\alpha_2$ at 0.1 and train the base 1.5B model. The training results indicate that when coefficients are fixed, the model performance degrades, demonstrating the effectiveness of the adaptive coefficients in our approach.

Table 3: Performance of different parameter settings.

| Complexity Thresholds $(n_1, n_2)$ | $\triangle acc$ | $\triangle Length$(%) |
|---|---|---|
| 20, 40 | 3.36 | -20.2% |
| 60, 100 | 2.67 | -19.7% |
| 40, 100 | 3.36 | -20.0% |
| **40, 80 (ours)** | **3.58** | **-21.7%** |

| Penalty Weights $(\lambda_1, \lambda_2, \lambda_3, \alpha)$ | $\triangle acc$ | $\triangle Length$(%) |
|---|---|---|
| **Different $\alpha$** | | |
| 0.1 | 3.48 | -9.5% |
| 0.3 | 2.26 | **-30.7%** |
| **0.2 (ours)** | **3.58** | -21.7% |
| **Different $\lambda$** | | |
| 0.025, 0.05, 0.075 | 3.39 | -18.2% |
| 0.1, 0.15, 0.2 | 3.08 | -18.8% |
| **0.05, 0.1, 0.15 (ours)** | **3.58** | **-21.7%** |

**The sensitivity analysis.** To explore the impact of the complexity thresholds $(n_1, n_2)$ and penalty weights $(\lambda_1, \lambda_2, \lambda_3, \alpha)$ on model performance, we evaluate various parameter configurations of ARLCP applied to the DeepSeek-R1-Distill-Qwen-1.5B model (trained for 100 steps). The results, summarized in Table 3, demonstrate that:

- **Impact of $n_1$ and $n_2$:** Increasing $n_1$ leads to a slight initial improvement in performance, followed by a decline; the best performance is achieved at $n_1 = 40$. In contrast, varying $n_2$ have a relatively minor effect on performance, with the optimal setting found at $n_2 = 80$.

- **Impact of** $\alpha$: A smaller $\alpha$ leads to a lower reduction in output length and lower accuracy, whereas a larger $\alpha$ yields greater length compression but at the expense of a more noticeable drop in accuracy. Our chosen $\alpha$ strikes a good balance between the length compression ratio and accuracy.
- **Impact of** $\lambda_1$, $\lambda_2$, $\lambda_3$: Furthermore, with $\alpha$ fixed, either increasing or decreasing the overall magnitude of the $\lambda$ parameters causes only minor fluctuations in performance, and the overall performance remains at a stable level.

Table 4: The performance of ARLCP on the out-of-distributeion benchmark MMLU. ARLCP achieves the best balance on MMLU, improving accuracy while significantly reducing response length compared to other baselines.

| Method | MMLU | | | |
|---|---|---|---|---|
| | Acc | Length | $\Delta$Acc | $\Delta$Length |
| *DeepSeek-R1-Distill-Qwen-7B* | | | | |
| Vanilla | 63.4 | 1257 | - | - |
| Nothinking | 51.2 | 128 | -12.2 | -89.8% |
| SFT | 62.8 | 1321 | -0.6 | +0.05% |
| AdaptThink | 63.6 | 856 | +0.2 | -31.9% |
| TLMRE | 63.9 | 872 | +0.5 | -30.6% |
| **ARLCP** | 64.1 | 742 | **+0.7** | **-41.0%** |

**The generalizability to different domains.** To evaluate cross-domain generalization, we assess ARLCP on MMLU, a challenging benchmark comprising 14k multiple-choice questions spanning 57 diverse subjects. As shown in Table 4, compared to the vanilla DeepSeek-R1-Distill-Qwen-7B, ARLCP reduces response length by 40% while achieving a 0.7% improvement in accuracy. These results demonstrate that ARLCP not only effectively transfers beyond mathematical reasoning to a broad range of open-domain tasks, but also consistently enhances reasoning efficiency across all question types. Importantly, the ability to maintain or even improve accuracy while significantly shortening responses highlights the robustness and generality of the proposed adaptive penalty mechanisms. This suggests that ARLCP is well-suited for practical deployment in diverse, real-world settings where both efficiency and accuracy are critical.

Table 5: Performance comparison of ARLCP on multiple reasoning benchmarks with Qwen3-1.7B.

| Method | AMC 2023 | | AIME 2024 | | AIME 2025 | | GSM8K | | MATH-500 | | Overall | |
|---|---|---|---|---|---|---|---|---|---|---|---|---|
| | Acc | Length | Acc | Length | Acc | Length | Acc | Length | Acc | Length | $\Delta$Acc | $\Delta$Length(%) |
| *Qwen3-1.7B* | | | | | | | | | | | | |
| Vanilla | 75.16 | 8484 | 38.75 | 13138 | 28.54 | 13287 | 90.67 | 2094 | 87.00 | 5096 | - | - |
| **ARLCP(ours)** | 79.69 | 5320 | 42.92 | 9466 | 32.92 | 9755 | 89.99 | 1057 | 89.60 | 2568 | **3.00** | **-38.19%** |

**The generalizability to different series models.** To evaluate the generalizability of our approach across model families, we extended our experiments to the Qwen3 series—using Qwen3-1.7B as a representative example—which lies outside the DeepSeek-R1-Distill-Qwen family. As shown in Table 5, ARLCP consistently improves accuracy while substantially reducing output length across nearly all benchmarks, demonstrating strong transferability beyond the original model family.

# 6 CONCLUSION

In this work, we first identify the phenomenon of over-reflection in reasoning models significantly impacting model efficiency and analyze its correlation with problem complexity. Based on this insight, we propose ARLCP, a dynamic reinforcement learning method that teaches LRMs to stop unnecessary reflection and enhance thinking efficiency through an adaptive reward strategy conditioned on problem complexity. Experiments on five mathematical reasoning benchmarks demonstrate that ARLCP reduces average response length while improving accuracy.

REPRODUCIBILITY STATEMENT

We have provided open-source code to reproduce all experiments described in this study (ARLCP). The datasets are publicly accessible, and all experimental procedures—including full implementation of model training, evaluation protocols, and reproducibility benchmarks—can be replicated using the provided resources.

ACKNOWLEDGMENT

This paper is mainly supported by the NSFC under Grants (No. 62402424). This work was supported by MYbank, Ant Group. Haobo Wang is also supported by the Fundamental Research Funds for the Central Universities (No. 226-2025-00085 & 226-2024-00145) and Zhejiang Provincial Universities (No. 226-2025-00065). This work is also supported by Fundamental and Interdisciplinary Disciplines Breakthrough Plan of the Ministry of Education of China.

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

## A  APPENDIX

### A.1  OPTIMIZING THE ADAPTIVE PENALTY WITH REINFORCEMENT LEARNING

The adaptive penalty in ARLCP (Equation 5) combines accuracy rewards with dynamic reflection/length penalties. However, autoregressive sampling during response generation renders this penalty non-differentiable. To address this, we employ Proximal Policy Optimization (PPO) (Schulman et al., 2017), a policy gradient method that maximizes the expected reward while constraining policy updates to avoid instability.

**PPO Objective with Density Ratio**   PPO optimizes the policy by maximizing the surrogate objective:

$$\min \left\{ f_\theta^k(y, x) \mathcal{A}(y^{<k}, x), \; clip_{1-\epsilon}^{1+\epsilon} \left[ f_\theta^k(y, x) \right] \mathcal{A}(y^{<k}, x) \right\}, \tag{7}$$

where the density ratio is defined as:

$$f_\theta^k(y, x) = \frac{\pi_\theta(y^k | x + y^{<k})}{\pi_{old}(y^k | x + y^{<k})}. \tag{8}$$

**RLOO Advantage Estimator**   Traditional PPO uses a value network to estimate advantages, which adds computational overhead for large language models. Instead, we adopt the REINFORCE Leave-One-Out (RLOO) estimator (Ahmadian et al., 2024), which computes advantages via Monte Carlo sampling. For a prompt $P$ with $m$ generated responses $O = [O^1, O^2 \ldots, O^m]$, the trajectory advantage is computed as:

$$\mathcal{A}(O^i, P) = \mathcal{R}(O^i, P) - \frac{1}{m-1} \sum_{j \neq i}^m \mathcal{R}(O^j, P), \tag{9}$$

where $\mathcal{R}(O^i, P)$ denotes the trajectory return. In ARLCP, the trajectory return corresponds directly to the reward function $r(o_i^k)$ from Equation 5, which integrates accuracy, reflection, and length penalties.

We approximate the token-level advantage by directly using the sequence-level advantage:

$$\mathcal{A}(y^{<t}, x) = \mathcal{A}(O^i, P). \tag{10}$$

This framework balances computational efficiency and performance by eliminating the value network while retaining robust policy updates through RLOO.

### A.2  THE DETAILS OF SELECTING RLOO

Although RLOO can produce high-variance gradient estimates, we choose it over the more popular GRPO due to a critical limitation of the latter: GRPO is prone to catastrophic collapse when combined with length-based rewards (Arora & Zanette, 2025; Dai et al., 2025). As shown in Table 6, our method also fails when using GRPO. To avoid this failure mode, we adapt RLOO and design our reward function accordingly.

To mitigate the theoretical variance of RLOO, we use standard strategies: large batch sizes (128) and multiple rollouts per prompt (16). Additionally, our penalty function provides regularization that further stabilizes training (see Equation 3). As shown in Figure 3, training remains stable throughout, confirming that RLOO is an effective choice for our setting—successfully avoiding the critical failure modes of GRPO.

Moreover, **RLOO is not essential** to our framework. To verify this, we implemented ARLCP using REINFORCE++ (Hu, 2025) as an alternative advantage estimator. As shown in Table 6, both the RLOO and REINFORCE++ variants achieve strong performance on DeepSeek-R1-Distill-Qwen-1.5B, demonstrating that our adaptive penalty mechanism is compatible with different policy

Table 6: Performance comparison between different RL methods.

| Method | Δ acc | Δ Length(%) |
|---|---|---|
| ARLCP with GRPO | -2.71 | -65.2% |
| ARLCP with REINFORCE++ | 4.87 | -50.4% |
| **ARLCP with RLOO (ours)** | 5.81 | -53.1% |

gradient algorithms. This flexibility enables ARLCP to effectively balance accuracy and conciseness across various RL advantage estimators, with the RLOO-based implementation yielding the best overall results.

### A.3 ANALYSIS OF REFLECTION-TOKEN COUNTS VERSUS PROBLEM COMPLEXITY ACROSS OTHER REASONING MODEL FAMILIES

To assess whether the correlation between reflection and problem complexity generalizes beyond a single model family, we examine the reflection behavior of GPT-OSS and Qwen3-Thinking across varying levels of problem complexity. As shown in Figure 5, both models exhibit the same trend observed in the DeepSeek-distilled Qwen models: the number of reflection tokens gradually increases with rising problem complexity. This result confirms that the relationship between reflection and problem complexity is not specific to one architecture, but rather holds across diverse reasoning models.

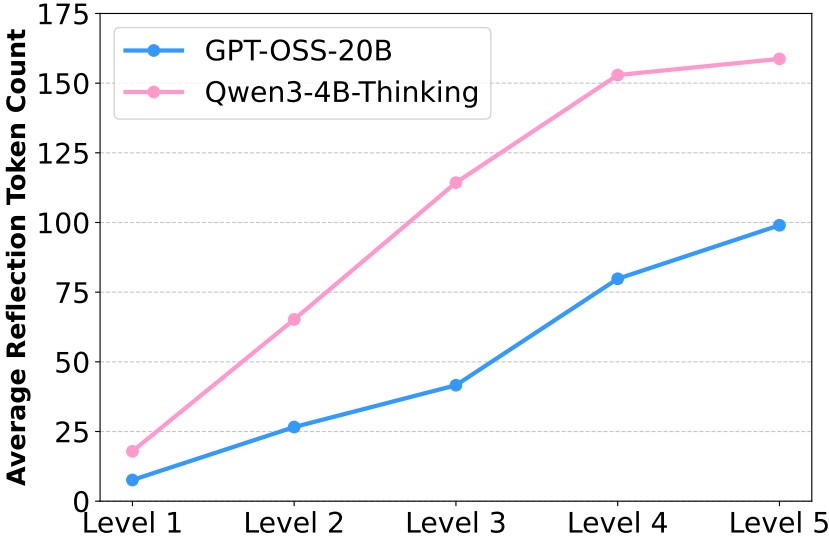

Figure 5: Average reflection token counts statistics of other models.

### A.4 REFLECTION TRIGGERS

We consider the following reflection-trigger keywords to calculate reflection token counts in our work, and their variants are also treated as reflection triggers to ensure comprehensive coverage. For instance, the variants of "wait" include "wait", "Wait", " wait", ".wait", and "WAIT". Similar expansions are applied to other reflection-trigger keywords to account for capitalization, spacing, and punctuation variations. These tokens are carefully curated to comprehensively capture reflection occurrences in the output, thereby ensuring both computational accuracy and analytical precision. The reflection-trigger keywords include:

> "wait", "alternatively", "hold on", "another thought", "verify", "think again", "but", "however", "alternative", "check", "double-check", "oh", "hmm".

## A.5 THE ANALYSIS OF FIXED REFLECTION TRIGGERS

Although our reflection keyword list is fixed, we emphasize that this approach is highly scalable within the current LRMs ecosystem. The list is derived through LLM-based reflection token recognition and statistical analysis of outputs from the DeepSeek-R1-Distill-Qwen models (1.5B and 7B).

To validate our claim, we analyzed reflection tokens in other LRM families and domains (such as finance), including Qwen3-4B-Thinking-2507, GPT-OSS-20B, and Fin-R1-7B Liu et al. (2025b). The most frequent reflection keywords for each model are shown as follows:

> **Reflection keywords in Qwen3-4B-Thinking-2507**
>
> "wait", "hold on", "maybe", "check", "think again", "another thought", "verify", "but", "hmm", "alternative".

> **Reflection keywords in GPT-OSS-20B**
>
> "maybe", "wait", "but", "alternative", "check", "however", "hmm", "double-check", "another thought".

> **Reflection keywords in Fin-R1-7B**
>
> "wait", "alternative", "but", "check", "however", "maybe", "perhaps", "hmm".

All three sets exhibit strong overlap with our predefined triggers. This cross-model consistency confirms that our reflection triggers captures universal reflection behaviors in modern LRMs. Consequently, using a fixed set does not hinder the generalizability of our method across different models or domains.

## A.6 PROMPT TEMPLATE IN EXPERIMENTS

Following prior work Zhang et al. (2025), we use the following prompt for all experiments: $question$. To ensure consistency across different settings, this base prompt is applied uniformly in every evaluation. Specifically, the complete prompt integrated with the models' templates (DeepSeek-R1-Distill-Qwen-1.5B and DeepSeek-R1-Distill-Qwen-7B) is constructed as follows:

> **Prompt Template for Reasoning in Experiments**
>
> $< |User| > \{question\} < |Assistant| >< think > \backslash n$

## A.7 THE DETAILS OF SELECTING COMPLEXITY THRESHOLDS ($n_1$, $n_2$)

We determined the complexity thresholds $n_1$ and $n_2$ through a pilot study on five benchmarks: GSM8K, MATH-500, AMC2023, AIME2024, and AIME2025. Using both DeepSeek-R1-Distill-Qwen-1.5B and 7B models, we measured the average number of reflection tokens per problem and found values to lie in the range $[0, 120]$, as shown in Figure 1. Based on this empirical distribution, we defined a three-tier complexity scheme:

- **Easy**: $0 \leq$ count $\leq 40$,
- **Moderate**: $40 <$ count $\leq 80$,
- **Hard**: count $> 80$.

This scheme is **model-agnostic**: to adapt it to a new model, one only needs to run inference on a representative set of problems across complexity levels, estimate its reflection token distribution, and align the thresholds accordingly.

- **Question (MATH-500)**: Define

$$p = \sum_{k=1}^{\infty} \frac{1}{k^2} \quad \text{and} \quad q = \sum_{k=1}^{\infty} \frac{1}{k^3}.$$

Find a way to write

$$\sum_{j=1}^{\infty} \sum_{k=1}^{\infty} \frac{1}{(j+k)^3}$$

in terms of $p$ and $q$.

* **Original Response**:

`<think>`Alright, so I have this problem here where I need to express a double sum in terms of two known series, p and q. Let me write down what I know: **[omited][...]**

But I don't think I need the numerical value here; it's just another constant.

Hmm, okay, so I have to manipulate this double sum somehow to relate it to p and q. **[omited][...]**

Wait, so for each fixed n, j goes from 1 to n-1, and k is n - j. **[omited][...]**

Wait, is that right? **[omited][...]** But wait, 1/n³ is independent of j, so the inner sum is just (n - 1) * 1/n³. **[...]** But I know that $\sum_{n=1}^{\infty} \frac{1}{n^2} = p$, so $\sum_{n=2}^{\infty} \frac{1}{n^2} = p - 1$. **[omited][...]**

Wait, let me double-check my steps to make sure I didn't make a mistake.

**[...][omit 2000 tokens]**

But wait, the integral **[omited][...]** Wait, yes, **[omited][...]** Wait, hold on, **[omited][...]** Wait, now I have two different results: **[omited][...]** Hmm, so which one is correct? Let me test with numerical computation.**[omited][...]**

**[...][omit 6000 tokens]**

Therefore, I think that the second method has a mistake, perhaps in the substitution or in handling the integrals. Alternatively, maybe I should accept that the first method is correct and S = p - q.

Therefore, the answer is S = p - q.

**Final Answer** $\boxed{p - q}$ `</think>`

**[omited][...]** Thus, the final answer is:

$$\boxed{p - q}$$

**(9713 tokens, 115 reflection tokens count)**

* **ARLCP's Response** (ours):

`<think>`Alright, so I have this problem here where I need to express a double sum in terms of two known sums, p and q. Let me write down what I know. **[omitted][...]** but maybe I don't need that exact value right now. Similarly, q is the sum from k=1 to infinity of $\frac{1}{k^3}$. That's known as Apéry's constant, but again, maybe **[omitted][...]** Hmm, **[omitted][...]** Wait, is that right? Let me check. **[omitted][...]** But since 1/n³ is independent of j, the inner sum is just adding 1/n³ (n - 1) times. **[omitted][...]** Therefore, the answer is p - q. Let me just double-check my steps to make sure I didn't make any mistakes. **[omitted][...]** **Final Answer** $\boxed{p - q}$ `</think>`

**[omitted][...]** Therefore, the final answer is:

$$\boxed{p - q}$$

**(1479 tokens, 8 reflection tokens count)**

Figure 6: For a math problem from MATH-500, DeepSeek-R1-Distill-Qwen-7B costs about 9000 tokens in thinking, which contains many unnecessary reflections. In contrast, our *ARLCP*-7B effectively reduces unnecessary reflections and produces a concise final solution.

## A.8    CASE STUDY

We present a case study of *ARLCP* in Figure 6 to demonstrate its capability in mitigating unnecessary reflections. As shown in the figure, when tackling challenging problems, the DeepSeek-R1-Distill-

Qwen-7B model generates significant token costs during reasoning, often accompanied by redundant and repetitive reflections. In contrast, our *ARLCP*-7B model dynamically stops unnecessary reflections through adaptive mechanisms, generating concise solutions while maintaining accuracy. This comparative analysis highlights the effectiveness of our approach in optimizing both computational efficiency and solution quality.

## A.9  LLM USAGE STATEMENT

This paper utilized Large Language Models (LLMs) as a general-purpose tool for language polishing and grammar refinement during writing. The role of LLMs is limited to enhancing the clarity and readability of the manuscript, including optimizing sentence structure and correcting errors in linguistic expression. No part of the core research methodology, data analysis, or theoretical contributions was generated or influenced by LLMs. The authors confirm that this disclosure complies with the conference's guidelines for transparency in LLM usage.

