# OpenReview forum: "Stop Unnecessary Reflection: Training LRMs for Efficient Reasoning with Adaptive Reflection and Length Coordinated Penalty"
_ICLR.cc/2026/Conference — ICLR 2026 Poster_

### Official Review · Reviewer_FW4W · 2025-10-30

**Soundness:** 3
**Presentation:** 4
**Contribution:** 3
**Rating:** 6
**Confidence:** 3

**Summary:**

This paper addresses the problem of computational inefficiency in Large Reasoning Models (LRMs) by identifying and targeting a specific failure mode over reflection, which is a pattern of repetitive and unproductive reasoning. The authors propose a novel reinforcement learning framework, Adaptive Reflection and Length Coordinated Penalty (ARLCP), designed to train models to produce more concise yet accurate reasoning paths.  It proposes a method based on empirical evidence linking over-reflection to incorrect answers, simultaneously improving reasoning accuracy while significantly reducing response length.

**Strengths:**

1. The paper offers an insightful framing of the efficiency problem by identifying over-reflection as a distinct, harmful behavior. The empirical analysis shows that incorrect answers correlate with higher reflection and token counts, providing a solid motivation for penalizing this behavior to improve both accuracy and efficiency.

2. The ARLCP framework is elegant and intuitive. Using the model’s own Reflection Token Count (RTC) as a real-time complexity proxy is a smart, self-contained design that removes the need for external classifiers. The adaptive penalty that balances reflection and response length based on complexity is a well-motivated and effective idea.

3. The experiments are thorough and convincing. ARLCP consistently outperforms strong baselines across multiple reasoning benchmarks and model sizes. The 1.5B model’s 5.8% accuracy gain with a 53% length reduction is a particularly impressive result.

4. Ablation studies and OOD tests (e.g., MMLU) demonstrate robustness and transferability. The results suggest that over-reflection is a general issue in reasoning models and that ARLCP provides a practical, broadly applicable mitigation.

**Weaknesses:**

1. The reliance on RTC as a proxy for complexity is a fragile heuristic. It can misclassify simple but confusing problems and, paradoxically, penalize the most reflective (and supposedly complex) cases most heavily—undermining the intended adaptivity.

2. The use of a manually curated list of ``reflection keywords'' is brittle and easily gamed. Models can paraphrase to avoid penalties, making this more of a short-term engineering fix than a scalable solution.

3. The paper introduces several crucial hyperparameters (e.g., $n_1, n_2, n_3, \lambda, \alpha$) that drive the penalty mechanism, but no sensitivity analysis is provided. This limits confidence in the method’s robustness and reproducibility.

4. The justification for using RLOO is incomplete. While alternatives are said to be unstable, the paper doesn’t discuss RLOO’s known high variance and potential training instability. Clarifying this trade-off would strengthen the technical argument.

**Questions:**

1. The paper states the goal is to "permit more extensive reasoning for complex ones," but the implementation penalizes reflection most heavily on problems with the highest RTC. Can you clarify this apparent contradiction between the stated goal and the reward mechanism?

2. The reflection penalty relies on a fixed keyword list. Have you analyzed whether the model learns to circumvent this penalty using synonyms, and how do you see this lexical approach scaling to other models with different linguistic patterns?

---

> ### Author Response · Authors · 2025-11-23
> **Response to Reviewer FW4W (1/2)**
>
> We sincerely thank the reviewer for acknowledging our work and recognizing its insightful problem framing, elegant self-contained design, strong empirical performance, and robust generalization.
>
> We apologize for the delay in our response; limited computational resources for LLM RL training mean that each experiment takes quite a while to complete. Below are our detailed responses to the questions.
>
> **Weakness 1 & Question 1: Non-contradictory learning objectives and penalty mechanisms**
>
> We emphasize that our learning goal is to **encourage models to think deeply to efficiently solve complex problems, rather than repeatedly experimenting and exploring with uncertainty**. The former's way of thinking can be considered as an expert type (efficient, linear, confident), which efficiently solves complex problems by identifying patterns and applying proficient strategies, thus exhibiting low RTC. The latter is a novice type (exploratory, iterative, uncertain), relying on trial and error due to a lack of professional knowledge, resulting in high RTC. Our ARLCP method encourages the LRM to reason like an expert so that we do not view high RTC as a sign of deep thinking when facing complex problems, but rather interpret it as a manifestation of cognitive inefficiency and therefore stop it, which is not contradictory to the learning goals we have set.
>
>
>
> **Weakness 2 & Question 2: Analysis of reward hacking of reflection tokens**
>
> We present evidence that reward hacking via synonym substitution is unlikely:
>
> 1. **Reduced Reflection Indicates Real Change:** If the model merely substituted keywords, the average reflection token count wouldn't have dropped significantly after ARLCP training. Instead, we observe a substantial decrease: a **53.05% reduction in response length** and a **+5.81% accuracy gain** for the 1.5B model. On AIME 2025, the average reflection count fell from **119.1 to 34.7 (a 71% reduction)**, demonstrating the penalty's effectiveness.
> 2. **RL Discourages Superficial Fixes:** ARLCP is a Reinforcement Learning method. It doesn't apply a filter; it trains the model to avoid inefficient reasoning *inherently*. Simply replacing a penalized keyword with a synonym doesn't make the reasoning more efficient and thus wouldn't be rewarded.
> 3. **Keyword Analysis Shows No Hacking:** We analyzed 30 responses from the ARLCP-7B model. The common reflection tokens found (`"wait", "but", "alternative", "hold on", "however", "hmm", "check"`) are the same as our original list, showing no widespread use of unpenalized synonyms.
>
> **Scalability to Models with Different Linguistic Patterns:**
>
> We argue that the keyword approach is highly scalable within the current LRM ecosystem.
>
> - **Generalization across "R1-style" Models:** Our method targets the reflection reasoning paradigm common to models like DeepSeek-R1 and OpenAI o1. The keywords capture universal "hesitation" signals (e.g., "wait", "hmm") during unproductive reflection, which are highly consistent across this model family.
>
> - **Low-Cost Adaptability:** The keyword list can be easily adapted for new models using low-cost LLM-judge techniques. As validation, we used an LLM to extract frequent reflection tokens from samples generated by `Qwen3-4B-Thinking-2507` and  `GPT-OSS-20B` on AIME2025. The resulting lists were substantially similar to ours, confirming the generalizability of our approach.
>
>   > **Qwen3-4B-Thinking-2507:** "wait", "hold on", "maybe", "check", "think again", "another thought", "verify", "but", "hmm", "alternative".
>
>   > **GPT-OSS-20B:** "maybe", "wait", "but", "alternative", "check", "however", "hmm", "double-check", "another thought".

---

> ### Author Response · Authors · 2025-11-23
> **Response to Reviewer FW4W (2/2)**
>
> **Weakness 3: Sensitivity to complexity thresholds and penalty factors.**
>
> As suggested by the reviewer, we investigate how the complexity thresholds ($n_1, n_2$) and penalty factors ($\lambda_1, \lambda_2, \lambda_3, \alpha$) affect the performance of our method on DeepSeek-R1-Distill-Qwen-1.5B after 100 training steps. The results are shown in Tables 1 and 2.
>
> **Table 1:  Performance of different  complexity thresholds**
>
> | $n_1, n_2 $         | $\Delta acc$ | $\Delta Length$ |
> | :------------------ | :----------: | :-------------: |
> | 20, 40              |     3.36     |     -20.2%      |
> | 60, 100             |     2.67     |     -19.7%      |
> | 40, 100             |     3.36     |     -20.0%      |
> | **40, 80  （ours)** |   **3.58**   |   **-21.7%**    |
>
> **Table 2:  Performance of different  reflection penalty weights**
>
> | Penalty Weights                        | $\Delta acc$ | $\Delta Length$ |
> | :------------------------------------- | :----------: | :-------------: |
> | **Different  $\alpha$**              |              |                 |
> | 0.1                                    |     3.48     |      -9.5%      |
> | 0.3                                    |     2.26     |   **-30.7%**    |
> | **0.2  (ours)**                        |   **3.58**   |     -21.7%      |
> | **Different  $\lambda$**             |              |                 |
> | 0.025,    0.05,       0.075            |     3.39     |     -18.2%      |
> | 0.1,        0.15,       0.2            |     3.08     |     -18.8%      |
> | **0.05,      0.1,        0.15 (ours)** |   **3.58**   |     **-21.7%**    |
>
> From Tables 1 and 2, we can draw the following conclusions:
>
> 1. **Impact of $n_1$ and $n_2$**:
>    Increasing  $n_1$ leads to a slight initial improvement in performance, followed by a decline; the best performance is achieved at $n_1$=40. In contrast, varying  $n_2$ has a relatively minor effect on performance, with the optimal setting found at $n_2$=80.
>
> 2. **Impact of $\alpha$**: A smaller $\alpha$ leads to a lower reduction in output length and lower accuracy, whereas a larger $\alpha$ yields greater length compression but at the expense of a more noticeable drop in accuracy. Our chosen $\alpha$ strikes a good balance between the length compression ratio and accuracy.
>
> 3. **Impact of $\lambda_1$, $\lambda_2$, $\lambda_3$**: Furthermore, with $\alpha$ fixed, either increasing or decreasing the overall magnitude of the $\lambda$ parameters causes only minor fluctuations in performance, and the overall performance remains at a stable level.
>
>
>
> **Weakness 4: Justification for RLOO**
>
> Our choice of REINFORCE Leave-One-Out (RLOO) was a deliberate trade-off.
>
> Although RLOO can produce high-variance gradient estimates, the popular alternative GRPO has a **critical flaw**. Since GRPO is prone to **collapse with length penalties** [1, 2], our primary algorithmic goal was to avoid this catastrophic failure, leading to our specific reward design.
>
> While RLOO has theoretical variance, we mitigated this using standard practices (large batch size of 128, 16 rollouts per prompt). Additionally, our penalty function provides regularization
> that further stabilizes training. As shown in our training curves (Figure 3 in the manuscript), the process was stable and convergent, confirming RLOO as an effective choice for our task that avoided the critical risks associated with GRPO.
>
> The supplementary justification for RLOO has been incorporated into the revised manuscript.
>
>
> **References**
>
> [1]  Stable Reinforcement Learning for Efficient Reasoning http://arxiv.org/abs/2505.18086
>
>
>
> [2] Training Language Models to Reason Efficiently http://arxiv.org/abs/2502.04463

---

### Official Review · Reviewer_dg2c · 2025-10-30

**Soundness:** 2
**Presentation:** 3
**Contribution:** 2
**Rating:** 4
**Confidence:** 3

**Summary:**

The authors observe that reasoning models often over-reflect, producing repetitive self-questioning and circular reasoning that lead to excessive token usage and computational overhead. These reflective behaviors are typically signaled by words such as wait, hmm, or check again. Their analysis shows that as problem complexity increases, models tend to generate more unnecessary reflection.
To address this, the authors propose estimating problem difficulty based on the frequency of reflection keywords. They introduce a reinforcement learning framework that adaptively applies both a reflection penalty and a length penalty. This approach encourages efficient reasoning by discouraging redundant thought patterns while maintaining problem-solving effectiveness.
Experiments using DeepSeek-distilled models demonstrate reductions in reasoning length and improved accuracy compared with existing methods.

**Strengths:**

1. The idea of estimating problem difficulty through the frequency of reflection keywords is novel and interesting.
2. The method is well-motivated, conceptually sound, and shows empirical effectiveness.
3. The writing is clear and the presentation is good.

**Weaknesses:**

1. The method may not generalize well to other model families. It assumes that reflection behaviors are expressed through specific keywords, which is a strong and model-dependent assumption. It would strengthen the paper if the authors extended the analysis of reflection-token counts versus problem difficulty to other reasoning model families (GPT-OSS, Qwen 3 Thinking, etc.).
2. All experiments are conducted on a single model family (DeepSeek-distilled Qwen), limiting the generality of the findings.
3. The paper's contributions are incremental. Prior studies [1,2] have already established the correlation between reflection tokens and reasoning performance, and existing works [3,4] have explored the use of length penalties to encourage concise reasoning.

References

[1] Demystifying Reasoning Dynamics with Mutual Information: Thinking Tokens are Information Peaks in LLM Reasoning

[2] s1: Simple test-time scaling

[3] ConciseRL: Conciseness-Guided Reinforcement Learning for Efficient Reasoning Models

[4] Walk Before You Run! Concise LLM Reasoning via Reinforcement Learning

**Questions:**

1. Does the proposed method also enable a non-reasoning model to acquire reasoning ability? The experiments only evaluate reductions in reasoning length for models that already possess reasoning capability.
2. The authors demonstrate that the adaptive penalty works effectively with RLOO. Would the approach generalize to other reinforcement learning techniques?

---

> ### Author Response · Authors · 2025-11-23
> **Response to Reviewer dg2c (1/2)**
>
> Thank you for your careful reading and insightful suggestions. We greatly appreciate your engagement with our work and your recognition of the novelty of our idea, the well-motivated design supported by strong empirical results, and the clarity and polish of our presentation.
>
> We apologize for the delay in our response; limited computational resources for LLM RL training mean that each experiment takes quite a while to complete. Below, we provide detailed responses to each of your points.
>
> **Weakness 1: Reflection keywords list**
>
> Based on our observation, our reflection keyword list can be generalized to different LRMs, such as GPT-OSS and Qwen-3-Thinking mentioned by the reviewer. Specifically, we obtained reflection keyword list of GPT-OSS and Qwen-3-Thinking:
>
> > Qwen3-4B-thinking: "**wait**", "**hold on**", "maybe", "**check**", "**think again**", "**another thought**", "**verify**", "**but**", "hmm", "**alternative**"
> >
> > GPT-OSS-20B: "maybe", "**wait**", "**but**", "**alternative**", "**check**", "**however**", "hmm", "**double-check**", "**another thought**"
>
> *Our reflection keyword list:*
>
> > "**wait**", "**alternatively**", "**hold on**", "**another thought**", "**verify**", "**think again**", "**but**", "**however**", "**alternative**", "**check**", "**double-check**", "oh".
>
> From the above lists, we observe that **the reflection keyword distribution of other LRMs is strongly overlap with our reflection keyword list**. This indicates that the reflection word list **will not affect the generalization of our method** on LRMs with different distributions and domains.
>
> As suggested by the reviewer, we conduct analysis of reflection-token counts versus problem complexity on GPT-OSS and Qwen-3-Thinking. The analysis results are reported below.
>
> **Table 1: Average reflection token counts statistics.**
>
> | model             | gsm8k | math500 | amc2023 | aime2024 | aime2025 |
> | ----------------- | ----- | ------- | ------- | -------- | -------- |
> | qwen3-4b-thinking | 17.9  | 65.2    | 114.3   | 152.9    | 158.7    |
> | gpt-oss-20b       | 7.6   | 26.6    | 41.6    | 79.8     | 99.0     |
>
> From Table 1, we observe that the analysis results on GPT-OSS and Qwen-3-Thinking **exhibit the same trend as those on the DeepSeek-distilled-Qwen models**: the number of reflection tokens gradually increases as problem complexity grows. We also include these results in Fig. 5 of the revised manuscript.
>
> **Weakness 2: Evaluation limited to a single model family**
>
> We have expanded our experiments to the Qwen-3 family LRMs (task Qwen-3-1.7B as an example), which is outside the DeepSeek-distilled family. And the results are shown below:
>
> **Table 2: The performance of ARLCP on Qwen-3-1.7B**
>
> | Method           | AMC2023      | AIME2024      | AIME2025      | GSM8K        | MATH-500     | Overall                     |
> | ---------------- | ------------ | ------------- | ------------- | ------------ | ------------ | --------------------------- |
> |                  | ACC / Len    | ACC / Len     | ACC / Len     | ACC / Len    | ACC / Len    | $\Delta ACC$ / $\Delta Len$ |
> | **Vanilla**      | 75.16 / 8484 | 38.75 / 13138 | 28.54 / 13287 | 90.67 / 2094 | 87.00 / 5096 | — / —                       |
> | **ARLCP (Ours)** | 79.69 / 5320 | 42.92 / 9466  | 32.92 / 9755  | 89.99 / 1057 | 89.60 / 2568 | **3.00**/ **-38.19%**                 |
>
> ARLCP consistently improves accuracy while significantly reducing output length across nearly all benchmarks, demonstrating its **effectiveness beyond the original model family**. These results have been incorporated into Section 5 of the revised manuscript.

---

> ### Author Response · Authors · 2025-11-23
> **Response to Reviewer dg2c (2/2)**
>
> **Weakness 3: Incremental contributions**
>
> We clarify that **our contribution is not incremental, but innovative**. While prior works study reflection tokens with training-free methods [1,2] or uses static length penalties for RL methods [3,4], they often offer limited efficiency gains, or fail to dynamically adapt response length to a problem’s complexity or mitigate a model’s over-reflection tendency.
>
> Unlike these methods, our key contribution lies in:
>
> 1. **First use of reflection tokens as RL signals**: Different from prior training-free methods [1,2] that do not chage the inherent capabilities of the models, we are the first to leverage reflection tokens as real-time reward signals in RL to actively shape model behavior.
> 2. **Adaptive, complexity-aware penalty mechanism**: Unlike [3,4], which apply static or uniform length penalties, our method dynamically adjusts reflection penalty based on problem complexity. This enables fine-grained control and always suppresses excessive reflection.
> 3. **Novel coordinated penalties design**: ARLCP jointly penalizes (a) raw output length and (b) reflection token counts. This dual signal more effectively stops over-reflection on complex tasks while maintaining efficiency on simpler ones—a capability absent in generic length-penalized methods.
>
> These key innovations **were not mentioned in prior methods**. Therefore, our contribution is **not incremental, but innovative**.
>
>
>
> **Question 1: Can ARLCP endow Non-Reasoning LLMs with reasoning ability?**
>
> We emphasize that ARLCP operates within an RL framework and therefore does not alter the fundamental nature of RL. As is well known, DeepSeek-R1 [5] enhances reasoning capabilities through RL (specifically GRPO), and our ARLCP similarly possesses this ability. To further illustrate this point, we evaluated ARLCP on the non-reasoning LLM Qwen2.5-7B-Instruct. Like GRPO, ARLCP further improves the reasoning performance of non-reasoning LLMs on mathematical tasks.
>
> **Table 3: The performance comparisons of Qwen2.5-7B-Instruct**
>
> | Method           | $\Delta acc$ | $\Delta Length$ |
> | ---------------- | ------------ | --------------- |
> | Vanilla          | —            | —               |
> | GRPO             | **+1.4**     | -2.0%           |
> | **ARLCP (Ours)** | +1.3         | **–20.8%**      |
>
>
>
> **Question 2: Generalization to other RL algorithms**
>
> Yes, **RLOO algorithm is not essential** to our framework. Our reward design is also compatible with the other group-based policy optimization algorithm. To demonstrate this, we conduct experiments using ARLCP with REINFORCE++. The results are shown in Table 4:
>
> **Table 4: The performance comparisons of different RL algorithms on DeepSeek-R1-Distill-Qwen-1.5B**
>
> | Method                 | $\Delta acc$ | $\Delta Length$ |
> | ---------------------- | ------------ | --------------- |
> | Vanilla                | —            | —               |
> | ARLCP-REINFORCE++      | 4.87         | –50.4%          |
> | **ARLCP- RLOO (ours)** | **5.81**     | **-53.1%**      |
>
> The strong performance confirms that our **adaptive penalty mechanism can be integrated into different RL algorithms**  to better balance accuracy and conciseness.
>
>
>
> **References**
>
> [1] Dynamic Early Exit in Reasoning Models http://arxiv.org/abs/2504.15895
>
>
>
> [2] Test-time Prompt Intervention  http://arxiv.org/abs/2508.02511
>
>
>
> [3] ConciseRL: Conciseness-Guided Reinforcement Learning for Efficient Reasoning Models https://arxiv.org/abs/2505.17250
>
>
>
> [4] Walk Before You Run! Concise LLM Reasoning via Reinforcement Learning https://arxiv.org/abs/2505.21178
>
>
>
> [5] DeepSeek-R1: Incentivizing Reasoning Capability in LLMs via Reinforcement Learning  https://arxiv.org/abs/2501.12948

---

> > ### Comment · Reviewer_dg2c · 2025-11-26
> >
> > I thank the authors for providing clear, in-depth explanations and additional experiments. My concerns are mostly addressed and I have revised the score accordingly.

---

> > > ### Author Response · Authors · 2025-11-26
> > >
> > > Thank you very much for your positive feedback! We are truly honored to have addressed most of your concerns. We sincerely appreciate the time and thoughtful effort you dedicated to reviewing our manuscript. Your insightful and constructive comments help us a lot for improving our work.

---

### Official Review · Reviewer_AdYW · 2025-10-30

**Soundness:** 3
**Presentation:** 3
**Contribution:** 2
**Rating:** 6
**Confidence:** 4

**Summary:**

This paper identifies the problem of **over-reflection** in Large Reasoning Models, where repetitive self-questioning causes inefficiency without accuracy gain. It proposes **ARLCP (Adaptive Reflection and Length Coordinated Penalty)**, a reinforcement learning method that dynamically adjusts reflection and length penalties based on problem complexity. Experiments on several math benchmarks show that ARLCP significantly reduces reasoning length while improving accuracy, achieving a better efficiency–accuracy balance than previous length-penalized approaches.

**Strengths:**

**Strengths**:
1. The paper is well-written and clearly structured, making it easy to follow.
2. It identifies a practical issue—over-reflection in large reasoning models—and provides a targeted solution.
3. The proposed ARLCP method is simple yet effective, adaptively balancing reflection and reasoning length.
4. The experiments on multiple benchmarks demonstrate clear improvements in both accuracy and efficiency.

**Weaknesses:**

typo line 448: analysis -> analyze

Weaknesses:
1. The method appears potentially sensitive to the hyperparameters n₁ and n₂. It would be helpful if the authors could analyze how these values affect stability and performance.
2. The paper lacks clear guidance or heuristics for selecting n₁ and n₂, which seem crucial to the method’s effectiveness.
3. The work assumes that longer reasoning tends to produce incorrect answers, yet the analysis mainly shows correlation rather than causation. It remains possible that more difficult problems naturally require longer reasoning.
4. The paper could better position itself relative to prior studies on difficulty dynamics in reinforcement learning for efficient reasoning, such as [1].

[1] Learn to Reason Efficiently with Adaptive Length-based Reward Shaping. https://arxiv.org/abs/2505.15612

**Questions:**

See weakness

---

> ### Author Response · Authors · 2025-11-23
> **Response to Reviewer AdYW (1/2)**
>
> Thank you for your constructive and thoughtful feedback. We sincerely appreciate the time and care you devoted to reviewing our work, and we are grateful for your recognition of its clarity-oriented presentation, problem-driven motivation, adaptive penalty design, and comprehensive empirical results.
>
> We apologize for the delay in our response; limited computational resources for LLM RL training mean that each experiment takes quite a while to complete. Below are our point-by-point responses to your concerns.
>
> The spelling mistake has been corrected.
>
> **Weakness 1: Sensitivity to complexity thresholds and penalty factors.**
>
> As suggested by the reviewer, we investigate how the complexity thresholds ($n_1, n_2$) and penalty factors ($\lambda_1, \lambda_2, \lambda_3, \alpha$) affect the performance of our method on DeepSeek-R1-Distill-Qwen-1.5B after 100 training steps. The results are shown in Tables 1 and 2.
>
> **Table 1:  Performance of different  complexity thresholds**
>
> | $n_1, n_2 $         | $\Delta acc$ | $\Delta Length$ |
> | :------------------ | :----------: | :-------------: |
> | 20, 40              |     3.36     |     -20.2%      |
> | 60, 100             |     2.67     |     -19.7%      |
> | 40, 100             |     3.36     |     -20.0%      |
> | **40, 80  （ours)** |   **3.58**   |   **-21.7%**    |
>
> **Table 2:  Performance of different  reflection penalty weights**
>
> | Penalty Weights                        | $\Delta acc$ | $\Delta Length$ |
> | :------------------------------------- | :----------: | :-------------: |
> | **Different  $\alpha$**              |              |                 |
> | 0.1                                    |     3.48     |      -9.5%      |
> | 0.3                                    |     2.26     |   **-30.7%**    |
> | **0.2  (ours)**                        |   **3.58**   |     -21.7%      |
> | **Different  $\lambda$**             |              |                 |
> | 0.025,    0.05,       0.075            |     3.39     |     -18.2%      |
> | 0.1,        0.15,       0.2            |     3.08     |     -18.8%      |
> | **0.05,      0.1,        0.15 (ours)** |   **3.58**   |     **-21.7%**      |
>
> From Tables 1 and 2, we can draw the following conclusions:
>
> 1. **Impact of $n_1$ and $n_2$**:
>    Increasing  $n_1$ leads to a slight initial improvement in performance, followed by a decline; the best performance is achieved at $n_1$=40. In contrast, varying  $n_2$ has a relatively minor effect on performance, with the optimal setting found at $n_2$=80.
>
> 2. **Impact of $\alpha$**: A smaller $\alpha$ leads to a lower reduction in output length and lower accuracy, whereas a larger $\alpha$ yields greater length compression but at the expense of a more noticeable drop in accuracy. Our chosen $\alpha$ strikes a good balance between the length compression ratio and accuracy.
>
> 3. **Impact of $\lambda_1$, $\lambda_2$, $\lambda_3$**: Furthermore, with $\alpha$ fixed, either increasing or decreasing the overall magnitude of the $\lambda$ parameters causes only minor fluctuations in performance, and the overall performance remains at a stable level.
>
> **Weakness 2: Details on selecting $n_1$ and $n_2$**
>
> We determined $n_1$ and $n_2$ via a pilot study on five benchmarks (GSM8K, MATH-500, AMC2023, AIME2024, AIME2025), measuring reflection token counts from both 1.5B and 7B models. As shown in Figure 1 of the paper, counts span [0, 120], leading us to define:
>
> - **Easy**: [0, 40]
> - **Moderate**: (40, 80]
> - **Hard**: >80
>
> This scheme is **model-agnostic**—new models only require an inference run to estimate their reflection distribution and set thresholds. These details are now included in the revised manuscript.
>
>
>
> **Weakness 3: Correlation vs. Causality in reasoning length**
>
> We clarify that we do not assume long reasoning causes incorrect answers; rather, it is **excessive reflection** that often leads to errors. Moreover, as shown on the right side of Fig. 2 in the manuscript, as reflection tokens increase, the accuracy drops sharply. This indicates that ineffective overthinking indeed provides no benefit to model performance.
>
> For difficult problems, we **expect the model to find the correct answer with a few reflections**. We achieve this by:
>
> 1. Penalizing excessive reflection behaviors, which correspond to redundant or unnecessary reasoning;
> 2. Adapting the penalty based on problem complexity, allowing longer reasoning for difficult problems while stopping excessive reflection.
>
> **This expectation does not contradict the need for longer reasoning chains.** As shown in Table 1 in the manuscript, ARLCP achieves 3% accuracy improvement on the most challenging benchmarks (AIME2024/2025), demonstrating it preserves essential reasoning and only suppresses harmful over-reflection—something naive length penalties cannot accomplish.

---

> ### Author Response · Authors · 2025-11-23
> **Response to Reviewer AdYW (2/2)**
>
> **Weakness 4: Positioning relative to prior work**
>
> We thank the reviewer for pointing us to the prior work **LASER** (*Learn to Reason Efficiently with Adaptive Length-based Reward Shaping*, arXiv:2505.15612). We have now included a direct comparison in Table 3.
>
> **Table 3: Performance Comparison with LASER on DeepSeek-R1-Distill-Qwen Models**
>
> | Method                            | AMC2023      | AIME2024      | AIME2025      | GSM8K        | MATH-500     | Overall                     |
> | :-------------------------------- | :----------- | :------------ | :------------ | :----------- | :----------- | --------------------------- |
> |                                   | ACC / Len    | ACC  / Len    | ACC  / Len    | ACC / Len    | ACC  / Len   | $\Delta ACC$ / $\Delta Len$ |
> | **DeepSeek-R1-Distill-Qwen-1.5B** |              |               |               |              |              |                             |
> | Vanilla                           | 66.72 / 7742 | 30.00 / 12256 | 21.40 / 12201 | 78.46 / 1009 | 80.20 / 4582 | — / —                       |
> | Laser                             | 75.94 / 3899 | 28.75 / 7629  | 25.42 / 7084  | 82.26 / 850  | 84.60 / 2365 | +4.04 / -38.69%             |
> | **ARLCP (Ours)**                  | 73.28 / 3037 | 34.17 / 5951  | 26.46 / 5196  | 87.34 / 654  | 84.60 / 1812 | **+5.81 / -53.05%**         |
> | **DeepSeek-R1-Distill-Qwen-7B**   |              |               |               |              |              |                             |
> | Vanilla                           | 87.50 / 5980 | 51.46 / 10547 | 35.00 / 11427 | 87.64 / 694  | 91.20 / 3673 | — / —                       |
> | Laser                             | 89.45 / 2894 | 56.45 / 5995  | 39.38 / 6410  | 86.73 / 818  | 91.00 / 1871 | +2.04 / -33.97%             |
> | **ARLCP (Ours)**                  | 89.69 / 3260 | 56.67 / 6795  | 39.58 / 7560  | 89.31 / 578  | 91.00 / 2086 | **+2.69 / -34.96%**         |
>
> From Table 3, ARLCP consistently outperforms LASER in **both accuracy gain and length reduction ratio**. And this baseline is now supplemented in Section 5 of our manuscript.
>
>
>
> We hope these clarifications address your concerns.

---

### Official Review · Reviewer_rUtM · 2025-11-01

**Soundness:** 3
**Presentation:** 3
**Contribution:** 2
**Rating:** 6
**Confidence:** 3

**Summary:**

This paper addresses the excessive and unnecessary reflection problem in Large Reasoning Models by proposing an adaptive reflection and length coordinated penalty (ARLCP). The method first estimates the problem's complexity by counting the Reflection Token Counts in a generated response, then assigns a penalty weight based on whether the problem is categorized as "simple," "moderate," or "hard." During RL training, a second penalty is applied to the overall response length to discourage general verbosity. The method achieves a superior efficiency-accuracy trade-off, simultaneously reducing length and improving accuracy.

**Strengths:**

- The work directly addresses the "over-reflection" in Large Reasoning Models (LRMs), a key issue that leads to high token consumption, computational overhead, and latency without improving accuracy.
- The core idea of an adaptive reflection penalty is a key innovation.
- The thorough analysis and ablation show that the method improves both efficiency and accuracy.

**Weaknesses:**

- The core "Adaptive Reflection Penalty" relies entirely on a manually curated list of "reflection-trigger keywords" like "wait", "hmm", "alternatively", etc. This relies on a specific model response style and is not generalizable to other distributions or domains.
- The method introduces several new and important hyperparameters: the complexity thresholds ($n_1=40, n_2=80$) and the penalty weights ($\lambda_1=0.05, \lambda_2=0.1, \lambda_3=0.15, \alpha=0.2$). The paper presents these as fixed values but does not include a sensitivity analysis.
- The method uses the output Reflection Token Count (RTC) as a proxy for the input problem's complexity. The paper itself shows that incorrect answers (i.e., failed reasoning) lead to higher reflection. This means the model might be penalizing "hard" for a problem that is actually "simple" but on which the model got stuck and "over-reflected." It's penalizing the symptom (high RTC) rather than the cause (intrinsic problem difficulty).

**Questions:**

- Have you tried directly using RLOO on the training dataset? I'm curious why this method achieves short outputs but still has the best performance.
- Did you face any instability during training? Does RLOO truly prove to be more stable?

---

> ### Author Response · Authors · 2025-11-23
> **Response to Reviewer rUtM（1/2）**
>
> Thank you very much for your valuable feedback, and for recognizing our clear over-reflection solution, the novelty of ARLCP, and the thorough analysis and ablation studies. We apologize for the delay in our response; limited computational resources for LLM RL training mean that each experiment takes quite a while to complete. We hope the following responses address your concerns.
>
> **Weakness 1: Non-manually constructed “reflection-trigger keywords”.**
>
>  We clarify that the list of “reflection-trigger keywords” is **automatically derived**, rather than manually curated. Specifically, it is obtained through LLM-based reflection recognition and statistical analysis of the outputs of the LRMs.
>  As suggested by the reviewer, we further conduct a statistical analysis of the reflection keyword distribution for other LRMs:
>
> *LRMs of other distributions:*
>
> > Qwen3-4B-thinking: "**wait**", "**hold on**", "maybe", "**check**", "**think again**", "**another thought**", "**verify**", "**but**", "hmm", "**alternative**"
> >
> > GPT-OSS-20B: "maybe", "**wait**", "**but**", "**alternative**", "**check**", "**however**", "hmm", "**double-check**", "**another thought**"
>
> *LRMs of other domains:*
>
> > Fin-R1-7B [1]: "**wait**", "**alternative**", "**but**", "**check**", "**however**", "maybe", "perhaps", "hmm"
>
> *Our reflection keyword list:*
>
> > "**wait**", "**alternatively**", "**hold on**", "**another thought**", "**verify**", "**think again**", "**but**", "**however**", "**alternative**", "**check**", "**double-check**", "oh".
>
> From the above lists, we observe that **the reflection keyword distribution of other LRMs is strongly overlap with our reflection keyword list**. This indicates that the reflection word list **will not affect the generalization of our method** on LRMs with different distributions and domains.
>
> **Weakness 2: Sensitivity to complexity thresholds and penalty factors.**
>
> As suggested by the reviewer, we investigate how the complexity thresholds ($n_1, n_2$) and penalty factors ($\lambda_1, \lambda_2, \lambda_3, \alpha$) affect the performance of our method on DeepSeek-R1-Distill-Qwen-1.5B after 100 training steps. The results are shown in Tables 1 and 2.
>
> **Table 1:  Performance of different  complexity thresholds**
>
> | $n_1, n_2 $         | $\Delta acc$ | $\Delta Length$ |
> | :------------------ | :----------: | :-------------: |
> | 20, 40              |     3.36     |     -20.2%      |
> | 60, 100             |     2.67     |     -19.7%      |
> | 40, 100             |     3.36     |     -20.0%      |
> | **40, 80  （ours)** |   **3.58**   |   **-21.7%**    |
>
> **Table 2:  Performance of different  reflection penalty weights**
>
> | Penalty Weights                        | $\Delta acc$ | $\Delta Length$ |
> | :------------------------------------- | :----------: | :-------------: |
> | **Different  $\alpha$**              |              |                 |
> | 0.1                                    |     3.48     |      -9.5%      |
> | 0.3                                    |     2.26     |   **-30.7%**    |
> | **0.2  (ours)**                        |   **3.58**   |     -21.7%      |
> | **Different  $\lambda$**             |              |                 |
> | 0.025,    0.05,       0.075            |     3.39     |     -18.2%      |
> | 0.1,        0.15,       0.2            |     3.08     |     -18.8%      |
> | **0.05,      0.1,        0.15 (ours)** |   **3.58**   |     **-21.7%**      |
>
> From Tables 1 and 2, we can draw the following conclusions:
>
> 1. **Impact of $n_1$ and $n_2$**:
>    Increasing  $n_1$ leads to a slight initial improvement in performance, followed by a decline; the best performance is achieved at $n_1$=40. In contrast, varying  $n_2$ has a relatively minor effect on performance, with the optimal setting found at $n_2$=80.
>
> 2. **Impact of $\alpha$**: A smaller $\alpha$ leads to a lower reduction in output length and lower accuracy, whereas a larger $\alpha$ yields greater length compression but at the expense of a more noticeable drop in accuracy. Our chosen $\alpha$ strikes a good balance between the length compression ratio and accuracy.
>
> 3. **Impact of $\lambda_1$, $\lambda_2$, $\lambda_3$**: Furthermore, with $\alpha$ fixed, either increasing or decreasing the overall magnitude of the $\lambda$ parameters causes only minor fluctuations in performance, and the overall performance remains at a stable level.

---

> ### Author Response · Authors · 2025-11-23
> **Response to Reviewer rUtM（2/2）**
>
> **Weakness 3: RTC (Reflection Token Counts) as a proxy for problem complexity.**
>
> We clarify **the problem complexity is model-aware**.  This complexity measures how difficult a problem appears to a given LRM, as reflected in its reasoning behavior, rather than as an inherent property of the problem. The same problem may trigger excessive reflection (high RTC) in a smaller model, indicating that it is difficult for that model, yet be solved directly (low RTC) by a larger model, indicating that it is easy for the larger one. Accordingly, we use RTC as a proxy for the model-aware complexity of a problem.
>
> Based on the above definition, the goal of our method is to **optimize reasoning behavior**. High RTC indicates inefficient reasoning—regardless of the problem’s inherent hardness. By penalizing this excessive reflective behavioral “symptom”, we encourage the LRM to adapt more efficient and effective reasoning strategies. As reported in our manuscript, this approach consistently improves both accuracy and efficiency across diverse benchmarks, confirming that stopping excessive reflection leads to better overall reasoning performance.
>
> These points have been clarified in the revised manuscript.
>
> **Question 1: What enables our RLOO variant to achieve both high accuracy and short outputs?**
>
> As shown in Table 3, **vanilla RLOO** (using only 0/1 correctness rewards) achieves only minor length reduction and **often produces long, repetitive reasoning traces with lower accuracy**, as the model gets stuck in unproductive reflection loops.
>
> Adding **standard length penalties slightly shortens outputs but fails to stop non-productive reflection**, and can even degrade accuracy by encouraging over-termination.
>
> **Our method** addresses these issues by **introducing adaptive penalties on both output length and reflection behaviors**. This directly **stops redundant thinking while maintaining—or even improving—correctness**.
>
> In short, the performance gains are mainly from our reflection-aware reward design, which jointly optimizes reasoning quality and efficiency.  A detailed case study is provided in the Appendix A.8 of the manuscript.
>
> **Table 3: Performance of different RL methods on DeepSeek-R1-Distill-Qwen-1.5B**
>
> | Method                  | $\Delta acc$ | $\Delta Length$ |
> | :---------------------- | :----------: | :-----------------: |
> | RLOO w/o penalty        |     5.74     |        -8.7%        |
> | ARLCP-GRPO              |    -2.71     |       -65.2%        |
> | ARLCP-reinforce++       |     4.87     |       -50.4%        |
> | **ARLCP- RLOO （ours)** |     5.81     |       -53.1%        |
>
> **Question 2: Training instability and RLOO stability.**
>
> Yes, we initially observed training instability when applying GRPO to our method: training collapsed early (after roughly 1/3 of all training steps), leading to sharp drops in both accuracy and token count (Table 3). This observation is consistent with prior work—[2] shows that GRPO collapses under length penalties, and [3] explains the underlying reasons—suggesting a **fundamental incompatibility between GRPO and length penalization**. Therefore, our final approach does not use GRPO.
>
> We therefore switched to the RLOO algorithm, which trained stably (see Fig. 3 in the manuscript) and ultimately achieved competitive performance. Moreover, when we applied our designed reward to another group-sampling algorithm, REINFORCE++, it also worked well, providing additional flexibility for practical deployment of our method.
>
> All supplementary content has been incorporated into the revised manuscript.
>
>
>
> **References:**
>
> [1] Fin-R1: A Large Language Model for Financial Reasoning through Reinforcement Learning https://arxiv.org/abs/2503.16252
>
> [2] Stable Reinforcement Learning for Efficient Reasoning [http://arxiv.org/abs/2505.18086]
>
> [3] Training Language Models to Reason Efficiently [http://arxiv.org/abs/2502.04463]

---

### Author Response · Authors · 2025-11-23
**General Response**

Dear Reviewers, AC, SAC, and PC,

We sincerely appreciate the time and effort you have devoted to reviewing our manuscript. We are especially grateful for the reviewers’ recognition of the novelty and soundness of our method, as well as the comprehensiveness and rigor of our experimental results. For clarity and convenience, we summarize below the key strengths highlighted by the reviewers, followed by our corresponding revisions and responses to their concerns.

(* We refer to Reviewer rUtM as R1, Reviewer AdYW as R2, Reviewer dg2c as R3, and Reviewer FW4W as R4)

According to R3's response, we have addressed **most of the reviewer's concerns**, and thus the score was updated to 6 on November 26.

------

### **Key strengths noted by the reviewers:**

**S1: Insightful Problem Identification**

[R1, R2, R4] The paper successfully identifies "over-reflection" as a critical issue, providing a *solid motivation* grounded in the finding that excessive reflection often correlates with incorrect answers.

**S2: Innovative and Elegant Design**

[R1, R3, R4] The ARLCP framework is recognized as *novel, simple, and intuitive*. Reviewers highlighted the smart use of internal reflection token counts (RTC) as a complexity proxy without needing external classifiers.

**S3: Strong Empirical Performance**

[All reviewers] The method demonstrates consistent and convincing improvements in *both accuracy and efficiency* across multiple benchmarks, achieving significant length reductions while maintaining or boosting performance.

**S4: Thorough Analysis**

[R1, R4] The experiments are thorough, including detailed ablation studies and out-of-distribution (OOD) tests that confirm the method's *robustness and transferability*.

**S5: Clear Presentation**

[R2, R3] The paper is *well-written and clearly structured*, making the concepts easy to follow and the contributions distinct.

------

### **Key responses and revisions to the comments**:

We have comprehensively responded to the reviewers’ feedback and revised the manuscript accordingly. The key responses and revisions are summarized below:

- [R1, R3, R4]: We clarify that the reflection keywords are automatically derived, and further analysis confirms **a significant overlap with other LRMs across diverse distributions and domains**, validating the **generalizability** of our method.  And we added the analysis of fixed reflection triggers in `Appendix A.5`.

- [R1, R2, R4]: As suggested by the reviewers, we supplemented the sensitivity analysis of complexity thresholds ($n_1, n_2$) and penalty factors ($\lambda_1, \lambda_2, \lambda_3, \alpha$) in `Section 5`.

- [R1]: We emphasize that problem complexity is **model-aware**, serving as a measure of how difficult a problem appears to a given LRM based on its reasoning patterns, as opposed to the problem's intrinsic difficulty. We have updated `Section 2 & 4` to clarify this point.

- [R1, R3, R4]: We have provided additional details regarding the selection of RLOO, including its justification, stability, and substitutability, in `Appendix A.2`.

- [R2]: We added details on the selection of $n_1$ and $n_2$, and discussed their generalizability in `Appendix A.7`.

- [R2]: We clarify that we do not assume long reasoning causes incorrect answers; rather, it is excessive reflection that often leads to errors.

- [R2]: Followed by the reviewer's suggestion, we supplemented a new baseline **Laser** in `Section 5`.

- [R3]: We have expanded our experiments in `Section 5` to include the Qwen-3 family (e.g., Qwen-3-1.7B), representing models outside the DeepSeek-distilled family.

- [R3]: We clarify that **our contribution is not incremental, but innovative**.

- [R3]: We emphasize that ARLCP preserves the fundamental nature of RL, evidenced by its ability to improve the mathematical performance of non-reasoning models like Qwen2.5-7B-Instruct, similar to GRPO.

- [R4]: We emphasize that our learning goal is to encourage models to think deeply to efficiently solve complex problems, rather than repeatedly experimenting and exploring with uncertainty.

- [R4]: We provide the analysis of reward hacking of reflection tokens.



All the revised contents are highlighted in blue.

---

### Meta-Review · Area_Chair_FvXU · 2025-12-29

**Summary:**

This submission proposes ARLCP to address LRM over-reflection via adaptive reflection and length-coordinated penalties. The authors fully addressed reviewers’ concerns. The work is well-motivated and empirically robust.

**Reviewer Concerns:**

Resolved concerns: reflection keyword generalizability, parameter sensitivity, complexity definition, RLOO details, and model diversity.

No unresolved critical issues.

**Reviewer Scores:**

Reviewer dg2c would update the score to 6 if able to participate fully in the discussion. All other reviewers initially gave 6 (above threshold).

---

### Decision · Program_Chairs · 2026-01-26

Accept (Poster)